# CNN-XG: A Hybrid Framework for sgRNA On-Target Prediction

**DOI:** 10.3390/biom12030409

**Published:** 2022-03-07

**Authors:** Bohao Li, Dongmei Ai, Xiuqin Liu

**Affiliations:** 1School of Mathematics and Physics, University of Science and Technology Beijing, Beijing 100083, China; s20200800@xs.ustb.edu.cn (B.L.); aidongmei@ustb.edu.cn (D.A.); 2Basic Experimental Center of Natural Science, University of Science and Technology Beijing, Beijing 100083, China

**Keywords:** Crispr/Cas9, sgRNA, on-target, deep learning, XGBoost

## Abstract

As the third generation gene editing technology, Crispr/Cas9 has a wide range of applications. The success of Crispr depends on the editing of the target gene via a functional complex of sgRNA and Cas9 proteins. Therefore, highly specific and high on-target cleavage efficiency sgRNA can make this process more accurate and efficient. Although there are already many sophisticated machine learning or deep learning models to predict the on-target cleavage efficiency of sgRNA, prediction accuracy remains to be improved. XGBoost is good at classification as the ensemble model could overcome the deficiency of a single classifier to classify, and we would like to improve the prediction efficiency for sgRNA on-target activity by introducing XGBoost into the model. We present a novel machine learning framework which combines a convolutional neural network (CNN) and XGBoost to predict sgRNA on-target knockout efficacy. Our framework, called CNN-XG, is mainly composed of two parts: a feature extractor CNN is used to automatically extract features from sequences and predictor XGBoost is applied to predict features extracted after convolution. Experiments on commonly used datasets show that CNN-XG performed significantly better than other existing frameworks in the predicted classification mode.

## 1. Introduction

The Crispr/Cas9 system is derived from the process by which phages infect bacteria. Crispr (clustered regularly interspaced short palindromic repeat) represents the sequence of short replies that are regularly spaced in clusters, with approximately the same length and specificity [1]. Crispr is a common immune system in bacteria used to fight viruses or exogenous DNA. The Crispr/Cas9 system directs the corresponding single guide RNA (sgRNA) recognition, positioning, fighting and cutting of target fragments of viral DNA by integrating invasive DNA fragments into the interval DNA. The recognition process is target recognition by the principle of complementary base pairing at a double-stranded target position of DNA with protospacer adjacent motif (PAM motif) [2,3].

Although Crispr/Cas9 relies on the principle of complementary base pairing for specific recognition, the Cas9 nucleases are tolerant to base matching between sgRNA and target DNA sequences. With the exception of cutting the target DNA duplex normally, sgRNA may also partially match with non-target DNA sequences with higher target homology, activating Cas9 to cut non-target sequences and producing off-target effects even though no mismatch exists at this time [4,5,6]. The off-target effects will seriously affect the practical application of Crispr. Effective evaluation of off-target and accurate prediction of on-target knockout efficacy of sgRNA has become the focus of Crispr/Cas9 system research.

Many models [7,8,9,10,11,12,13,14,15,16,17] have been developed to predict sgRNA on-target knockout efficacy with their own characteristics. From the beginning of the application of machine learning models in this field, sgRNA Designer [7] constructed a model for predicting sgRNA on-target cleavage efficacy with the help of machine learning algorithms such as SVM and random forest, characterized by biological sequence information such as sgRNA dinucleotide and GC content. Then, deep learning algorithms were gradually applied. DeepCrispr [8] introduced sgRNA sequence information and epigenetic information as characteristics for “one-hot” coding and used a CNN to build a framework for model prediction. In addition, they automated the whole feature identification procedure purely based on the available training data and the learning model and generated the feature saliency map for sgRNA on-target site prediction based on the existing training data to identify the preference for purine mismatches. DeepCpf1 [9] was characterized by sgRNA sequence and chromatin accessibility and built a prediction model based on a CNN. With “Transformer” showing good results in the field of natural language processing (NLP), the model based on an attention mechanism has been noticed. AttnToCrispr_CNN [10] encoded each position into a vector in a positional embedding layer as part of the characteristics and integrated features learned from advanced transformer-based deep neural network and CNN to build a prediction model. CnnCrispr [11] combined biLSTM and a CNN into a final predictive model. CNN-SVR [12] combined two hybrid architectures, CNN and SVR, for sgRNA on-target prediction. The success of the CNN-SVR model [12] inspired us to extend the application of a hybrid model for CRISPR/Cas9 sgRNA efficacy prediction. In areas such as image classification, a CNN is an efficient neural network learning model, whose convolution kernel plays an important role in feature extraction. A CNN allows [18] computational models that consist of multiple processing layers to learn representations of features with multiple levels of abstraction. The layers of features are learned from data through a generic learning process instead of human engineers. CNNs [19] are multi-layer architectures where the successive layers are designed to progressively learn higher-level features. Although a CNN is considered as one of the most powerful and effective feature extraction mechanisms, the traditional classifier layer of a CNN cannot fully capture the information of the extracted features. Single classifier cannot perform well when faced with complex data features, in this case, ensemble learning combines multiple classifiers together and often achieves good results. Chen [20] proposed an advanced gradient enhancement algorithm, extreme gradient boosted tree (XGBoost), which achieved good results in the Kaggle data competition. XGBoost has been widely used in image classification [21,22] with good performance. Ren et al. [23] proposed a CNN and XGBoost based image classification method. In this model, a CNN is used to obtain image features from the input and XGBoost as a recognizer to produce results to provide more accurate output.

Although there has been some progress and breakthroughs in the prediction of sgRNA on-target cleavage efficacy and many model improvements have been made in recent years, the accuracy and generalization ability of the model still need to be further improved. Here, we present a hybrid machine learning framework called CNN-XG. As we have described before, a CNN is not always the best choice for classification, instead, the advanced ensemble model XGBoost can overcome the deficiency of a single classifier to classify features and thus exhibit good predictive performance. We would like to improve the prediction performance of the model by introducing XGBoost. The idea of CNN-XG is to train a dedicated CNN network to extract initial sgRNA genetic information and epigenetic information and then provide effective features obtained through convolution and pooling to XGBoost for prediction and evaluation. First, we trained the CNN model on a benchmark dataset for model pre-training, aiming at model selection and parameters tuning. Second, the features extracted from data via the CNN were input into XGBoost for training and evaluation. Third, the trained CNN-XG was used to test the independent dataset. Results show that CNN-XG surpassed the state-of-the-art tools in most tests.

## 2. Materials and Methods

### 2.1. Data Sources

#### 2.1.1. Benchmark Dataset

Based on previous studies by Doench JG et al. [7,24], Guohui Chuai et al. [8] adopted data augmentation strategy such as that used in image data processing and obtained approximately 200,000 non-redundant sgRNAs with biologically meaningful knockout efficacies for the training process. By using this data as a benchmark dataset, we completed the pretraining of a CNN.

#### 2.1.2. Four Cell Line Independent Datasets

In order to evaluate the performance of CNN-XG, we used a total of 1071 sgRNAs from four experimental, validated, sgRNA on-target cleavage efficacy, independent human datasets, HCT116, HEK293T, HELA and HL60, which were integrated and processed by Chuai et al. in DeepCrispr [8]. Each piece of data in the dataset contains a 23-nt sgRNA sequence, four kinds of corresponding symbolic epigenetic features, as well as numerical and binary cleavage efficacy. Epigenetic features were obtained from ENCODE [25], including CTCF binding information from the ChIP-Seq assay, H3K4me3 position information from the ChIP-Seq assay, chromatin-opening information from the DNase-Seq assay, and DNA methylation information from the RRBS assay.

#### 2.1.3. Three Independent Datasets for Generalization Test

The dataset was carefully selected from previously published literature [26,27] by Qiao Liu et al. [10]. Approximately 105,000 sgRNAs, 74,000 sgRNAs and 74,000 sgRNAs were studied in K562, A549 and NB4 cell lines, respectively. In these experiments, the log2 fold change (log2fc) of sgRNA counts between before and several days after treatment with the CRISPR-Cas9 system was calculated and normalized for each sgRNA. The normalized log2fc was used for on-target efficiency prediction.

#### 2.1.4. SpCas9 Variant Datasets

Streptococcus pyogenes Cas9 (SpCas9) has been widely used for genome editing, partly owing to its high activity and relatively broad PAM compatibilities. Wang et al. [28] performed a genome-scale screen to measure sgRNA activity for two highly specific SpCas9 variants (eSpCas9 (1.1) and SpCas9-HF1) and wild-type SpCas9 (WT-SpCas9) in human cells and obtained indel rates of over 50,000 sgRNAs for each nuclease, covering ~20,000 genes. After removing the non-edited sequences, we obtained indel rates of 58,616, 56,887 and 55,603 sgRNAs for these three nucleases; we called them ESP, HF and WT.

Kim et al. [29] extensively compared the activity, specificity and PAM compatibilities of 13 SpCas9 variants, and we selected three of the thirteen SpCas9 variants data: Sniper-Cas9, SpCas9-NG, and xCas9. Target sequences associated with low indel frequency were excluded from this study. After removing the redundancy, the number of datasets for Sniper-Cas9, SpCas9-NG and xCas9 was 37,794, 30,585 and 37,738. We called them Sniper, SpCas9 and xCas9.

### 2.2. Design of CNN-XG

We proposed a network combining CNN and XGBoost called CNN-XG to provide a deep learning method for CRISPR/Cas9 sgRNA on-target activity prediction. As shown in Figure 1a–d, CNN-XG receives a 23-nt sgRNA sequence and four “A2013N” symbolic epigenetic sequences with a length of 23 as inputs. Then, the CNN part outputs the features learned from sequences as the input of the XGBoost predictor, and it produces a regression score of sgRNA on-target cleavage efficacy. Compared to machine learning-based approaches that rely heavily on manual features, CNN-XG could get rid of the reliance on manual feature engineering.

### 2.3. Sequence Encoding

We used the coding method commonly used in machine learning: one-hot coding to encode sgRNA information and epigenetic information, which takes into account not only the nucleotide information of sgRNA, but also the epigenetic information of the corresponding location. There are four base pairs, A, G, C and T, in total. For a sgRNA sequence of 1 × 23, we can use four binary channels, A channel, G channel, C channel and T channel, to encode the base information. If the corresponding position for each channel is 1, it represents the presence of the corresponding nucleotide, otherwise, it is represented by 0. Taking the T-channel as an example, the presence of the nucleotide T at a particular base pair position was denoted by 1 and the absence of the nucleotide T was represented by 0. Consequently, each sgRNA was expressed by a 4 × 23 matrix, where 23 was the length of the sgRNA sequence (Figure 1a).

We encoded four pieces of epigenetic information in the same way. Each channel used 1 to indicate the existence of this information, 0 to indicate that the information does not exist, and the epigenetic characteristics corresponding to 23 locations were encoded into a matrix of 4 × 23. The matrices of sgRNA and epigenetic characteristics were then fed into convolutional neural networks for training.

### 2.4. CNN Model Establishing

CNN-XG is organized in a sequential layer-by-layer structure, where the CNN model plays a key role in extracting deep features of sgRNA sequence and its corresponding epigenetic information. As shown in Figure 1e, the CNN network contains a sgRNA sub-network for extracting features from sgRNA and an epigenetic stream sub-network for extracting features from the four epigenetic features. The two sub-networks are structurally identical and include three one-dimensional (1D) convolutional layers, three max-pooling layers, one flattening layer and four fully connected layers.

Taking the sgRNA part as an example, it accepts the 4 × 23 binary matrix as input. The first layer is a 1D convolution layer, which is applied to extract the sgRNA features using 64 convolution kernels of size 3. A rectified linear unit (ReLU) is subsequently used as the activation function to the convolution outputs. The max-pooling layer, applying a filter with window size 2 to the previous layers, is used to reduce the number of parameters. The remaining two convolution layers use 128 convolution kernels of size 3 and 256 convolution kernels of size 3. The structures of the following max-pooling layer are consistent with the first pooling layer, respectively. Outputs of the last pooling layer are joined together into one vector via flattening. After that, the features are followed by four fully connected layers with the sizes of 256, 128, 64 and 32, respectively. The features of the fourth fully connected layer from both sgRNA and epigenetic branches are concatenated by the “concatenate” operator. The outputs of the concatenation layer are input into the last fully connected layer of the merged CNN network. The final output layer consists of one neuron corresponding to the predicted score. Dropout is applied for the model regularization to avoid overfitting and the drop rate is determined to be 0.2.

### 2.5. Pre-Training of CNN

The architecture we proposed, called CNN-XG, is a predictive algorithm for predicting sgRNA on-target knockout efficacy. Before training the model, we first complete the pre-training and parameter determination of the CNN part. During the pretraining process, we randomly assigned the samples of the dataset with 80% of samples for training and 20% of samples for testing with 5-fold cross-validation in the training process. The training set is randomly divided into five equal parts. In each training, one part was regarded as the testing dataset, while the remaining four parts were taken as the training dataset. Cross-validation allowed each dataset to be included in training, which contributes to effectively avoiding overfitting and guaranteeing the accuracy of CNN-XG. We select a set of weights that have the least loss function value on the validation set during the training process to save as the weight parameter for the CNN section.

### 2.6. Feature Representation Optimization

The initial features were transformed by the CNN, and when the outputs of the two branches of sgRNA and epigenetics were connected together via a flattened layer, we obtained the 64-dimensional features. Considering too many features might cause redundancy, and part of the features did not play a role in predicting the results; we adopted a strategy of optimizing features. We used random forests for feature screening. With Scikit-learn, we trained a random forest model with 1000 decision trees and entered 64-dimensional features into a random forest model. We obtained the importance score of each feature. Figure 2 shows the importance score of all the features, and we put the top 15 features into the the XGBoost classifier for training to get the final model.

### 2.7. XGBoost Model Training

We implemented the proposed methods in Python 3.9.6 and Keras library 2.4.3 with a Tensorflow (2.4.1) backend. The training and testing processes were performed on a desktop computer with Intel(R) Xeon(R) CPU E5-2650 v4 @ 2.20GHz, Ubuntu 18.04.2 LTS (GNU/Linux 4.15.0-162-generic x86_64) and 62.7 GB RAM. Two NVIDIA Corporation GP102 11.7 GB of memory per GPU were used to accelerate the training and testing process.

We output the random forest-optimized features to the XGBoost classifier [30]. Grid search was adopted to tune the hyperparameters of the proposed architectures: the number of the trees, the depth of the tree and the minimum weight of the leaf node. After optimization, the hyperparameters were as follows: the number of the trees:500; the depth of the tree:6; the minimum weight of the leaf node:1. We trained the framework of CNN-XG with the above hyperparameters.

## 3. Results

### 3.1. Overview of CNN-XG Model Architecture

For the prediction of on-target cleavage efficiency, we built CNN-XG (Figure 1). The framework is divided into two parts. The first part is a CNN as the front end of the model for learning and extracting sgRNA genetic information and epigenetic information. The second part is XG-Boost as the back end of the model, which is used to predict the on-target cleavage efficiency of sgRNA. The overall process of the CNN-XG framework is shown in Figure 1a–d. First, the sgRNA sequence and epigenetic sequence are converted into two 4 × 23 binary matrices via one-hot encoding, and then, the encoded sgRNA and epigenetic sequence are fed into the CNN and RF for feature extraction, and XG-Boost is trained based on the extracted characteristics. Ultimately, the well-trained XG-Boost model assigns a prediction score for testing sgRNA.

### 3.2. Comparison with CNN Model and XGBoost Model

To verify the feasibility of our approach, we compared our CNN-XG with the CNN model and the XGBoost model separately on each of the four cell line datasets. The current strategy of training is to use a 10-fold cross-validation approach.

For the CNN, we built a new framework with more complex structures to achieve the best performance of the single CNN. Two parameters, the activation function of the predictor and batch size during training, were optimized by using a grid search approach. For XGBoost, we merged the one-hot encoding vector of each base as the feature vector. Similar to sequence coding (see the “2.3. Sequence Encoding” Section), each base in the sequence can be encoded as one of the four one-hot vectors (1,0,0,0), (0,1,0,0), (0,0,1,0) and (0,0,0,1), and then, we concatenate each one-hot vector in order, so that we can get a 1 × 92 feature vector. Table 1 summarizes the results of 10 rounds of cross-validation tests of the final CNN model, XGBoost model and CNN-XG in both classification and regression modes. CNN-XG gets the best performance in both AUROC values and spearman coefficients. These results indicate that CNN-XG is more predictive than CNN or XGBoost working alone for sgRNA on-target activity, further confirming the feasibility and effectiveness of the combination of CNN and XGBoost, showing the superiority of the hybrid model.

### 3.3. Model Comparison with State-of-the-Art Methods

We selected five sgRNA on-target cleavage efficiency prediction models to compare with CNN-XG: sgRNA Designer [7], Deepcrispr [8], attnToCrispr_CNN [10], CNN-SVR [12] and SSC [31].

To make a more comprehensive comparison with the above model, we used the data from Deepcrispr, which includes four types of cell lines: hct116 [32], hek293t [7],hela [32] and hl60 [33]. In addition, we used the data from CRISPOR, which contains an experimentally proven target cleavage efficiency of 15,000 sgRNA data in 1071 genes. In our study, we made rigorous comparisons with other models, including classification and regression modes.

To make a strict comparison with other models, we tested the performance of different models in classification mode and regression mode, under “leave one out” and “10-fold cross-validation” test methods, respectively.

#### 3.3.1. Testing Scenario 1—Classification Schema

In this test, for 16,749 sgRNA sequences from four cell types and their on-target cleavage efficiency, 10% of the data from each cell type was randomly extracted as independent testing sets, while the remaining 90% of the data from each cell type was combined together for model training and parameter tuning during the 10-fold cross-validation process. CNN-XG is compared with the above five prediction models with the same classification testing mode. The comparison was evaluated using the values from an area under the receiver operating characteristic curve (AUROC). The results (Table 2, Figure 3a) showed that the AUROC values were 0.9732, 0.9905, 0.9714 and 0.9706 higher, respectively, than the other models in the data of HCT116, HEK293T, HELA and HL60.

#### 3.3.2. Testing Scenario 2—Classification Schema

In this test, we further tested the generalization ability of CNN-XG in new cell types. Then, CNN-XG was trained in a “leave one cell type out” way, each time leaving one cell type out and testing it, while using the training data combined from three cell types. Taking HCT116 as an example, we combined the training data of HEK293T, HELA and HL60 cell types and tested them on the test data of HC116 to compare the generalization abilities of different models in this way. The results (Table 3, Figure 3d) showed that when we left HCT116, HEK293T, HELA and HL60 out in turn, the AUROC values were 0.9721, 0.9695, 0.992 and 0.9708. CNN-XG shows good model classification performance in all test sets.

#### 3.3.3. Testing Scenario 3—Regression Schema

In this test, the whole comparison was performed in a similar way as in testing scenario 1, except that the model was trained in a regression way, and we selected the corresponding on-target cleavage efficacy values in regression mode as the label values. We completed the experiment with the same training set and test set.

We chose Spearman coefficient and Pearson coefficient as metrics for evaluation, respectively. Because models such as DeepCrispr are encapsulated and it is difficult to fully reproduce the training details, we could not obtain their Pearson coefficient during this training scenario. We only compared with the two remaining models (CNN-SVR attnToCrispr_CNN). The results of the two comparisons (Table 4 and Table 5, Figure 3b,c) show that CNN-XG does not necessarily have the best performance on the four datasets. However, in general, the hybrid models (CNN-XG and CNN-SVR) have a much more stable performance compared to the other models, consistently performing better in four cell lines without extremely poor performance as with other models. As a summary, we concluded that CNN-XG performed generally well for sgRNA on-target knockout efficacy prediction in regression mode.

#### 3.3.4. Testing Scenario 4—Regression Schema

We further tested the regression-based CNN-XG in a leave one cell type out way to investigate its generalization ability in new cell types, similar to test scenario 2, but we used regression label values and training methods. The results (Table 6, Figure 3e) showed that the Spearman coefficient of CNN-XG was higher than that of other models in HEK293T, HELA and HL60 and smaller than DeepCrispr and attnToCrispr_CNN in HCT116.

### 3.4. Generalization Ability Test on Independent Datasets

To better compare the generalization abilities of CNN-XG and other existing models for new data, we retested data from three cell lines, K562, A549, and NB4, collected from the CRISPR-Cas9 experiment [26,27]. In this test, we normalized the log2 fold change (log2fc) of sgRNAs as labels in regression mode for training and analysis. We tested the trained models in three cell lines data to compare their generalization ability, and the Pearson coefficient and p-value comparison results are shown in Table 6 and Figure 3f. We can see that CNN-XG also outperformed other models in new datasets. It appears that all the correlation coefficients are small in Table 7. After a further significance test, we found that there were significant correlations between the predicted results and the real values (*p*-value < 0.001) for the three methods on all test datasets. Furthermore, the *p*-value of CNN-XG is much smaller than that of the other two methods on the three test datasets. This result shows that CNN-XG has better generalization ability than other methods on independent datasets.

### 3.5. Comparison in SpCas9 Variant Data

SpCas9 variants have been developed to improve an enzyme’s specificity or to alter or broaden its protospacer adjacent motif (PAM) compatibility, but selecting the optimal variant for a given target sequence and application remains difficult. The differences between the variants and the PAM create heterogeneity of the data, and the previous model may not be applicable to the new variant data. Therefore, it is necessary for us to test the adaptability of CNN-XG on the new data. There are already many models that have been trained using SpCas9 variants data. Here, we use the same strategy that CRISPR-ONT adopts for on-target comparison. Each dataset was randomly divided into a training dataset and independent test dataset with the proportion of 85%:15%. The training process was performed under 10-fold cross-validation on each training dataset. With this testing scheme, we compared CNN-XG with CRISPR-ONT [34], DeepHF [28], C-RNNCrispr [35], DeepCas9[15] and DeepSpCas9 [29]. We retrieved sgRNA prediction efficiency scores of other algorithms from a CRSPR-ONT [34] study.

The comparison results are shown in Figure 4. We can see that the prediction performance of CNN-XG is similar to CRISPR-ONT, which is the previously published best performing method, with an average increase of 1.82% compared with the second best, DeepCas9. Experiments on various SpCas9 variants datasets demonstrate the effectiveness of our proposed CNN-XG for CRISPR/Cas9 sgRNA on-target activity prediction, which still performs well in the new data. Therefore, we conclude that CNN-XG is competitive against existing methods.

We further compared the running time of a CNN integrating XGBoost-based XGBoost, a CNN integrating attention-based CRISPR-ONT, the CNN-RNN-based C-RNNCrispr and the pure RNN-based DeepHF. Table 8 reports the training time for each predictor on five datasets over 10-fold cross-validation. We observe that CNN-XG has a significant advantage in terms of runtime. More specifically, we trained CNN-XG using two NVIDIA Corporation GP102 GPUs, spending about 1 h for these datasets. Because of its internal structure, CNN-XG does not need a lot of computational resources. Combined with the results in Figure 4, CNN-XG can achieve good performance with little time and resources for sgRNA efficiency prediction.

## 4. Discussion

The prediction of sgRNA on-target cleavage efficacy is critical to the development of the Crispr/Cas9 system. In this paper, we propose a hybrid framework called CNN-XG for predicting the on-target cleavage efficacy in the Crisp/Cas9 system. We use a CNN as the front end for model training and feature extraction and XGBoost as the back end for on-target cleavage efficacy prediction. Compared with other existing models, CNN-XG has demonstrated excellent performance in classification mode and achieved relatively good performance in generalization in regression mode.

In addition, during the comparison process, we focused on DeepCrispr, CNN-SVR and attnToCrispr_CNN. DeepCrispr is a representative deep learning model in recent years, and CNN-SVR is the latest integrated model introduced. They both use four cell line data, HCT116, HEK293T, HL60 and HELA, to train. attnToCrispr_CNN is the first model to introduce “transformer”, commonly used in the field of natural language processing. The result of the comparison is that CNN-XG has great advantages in classification mode, both in 10-fold cross-validation (Table 1) and in the test of the generalization ability of the model (Table 2). Compared with several other models, CNN-XG has shown obvious advantages probably because XGBoost is an ensemble learning algorithm based on the idea of boosting, integrating many underfitting weak learners. Kearns and Valiant [36] showed that weak classifiers can generate high precision estimates via integration, as long as data is sufficient.

Generalization is a drawback of deep learning-based methods for CRISPR/Cas9 activity and specificity prediction, namely, the model performs well only in specific datasets, but not in unseen datasets. We completed a number of tests regarding the generalization of the model, whether using a “leave one cell type out” approach (Table 3 and Table 6) or testing the trained CNN-XG directly on new cell line data (Table 7) or comparing it with recent methods in the SpCas9 variant dataset from multiple perspectives (Figure 4). The results show that CNN-XG has very good generalization ability, and this is very meaningful for the practical application of the model. In addition, the improvement of many models in accuracy comes at the expense of the increased computational time. It is necessary to compare the computation time of the models. We ran our experiments on an Ubuntu server with two NVIDIA Corporation GP102 GPUs with 11.7 GB of memory per GPU. CNN-XG took about 10 min on average to train over tens of thousands of data per dataset, achieving a performance close to that of the best models. Time is an important factor to consider when models are used in large-scale practical applications in the clinic or to analyze the efficiency of sgRNA on-target cleavage. From this perspective, CNN-XG has a big competitive advantage. Some improvements are expected in the future: (1) we will explore more on the simplified model, as we find that the structure of epigenetic information running parallel to sgRNA information is not particularly obvious for the improvement of prediction results, thus, we will try more ways to embed epigenetic information. (2) More complex deep learning models and frameworks in the field of natural language processing await future exploration, which is expected to further improve the performance of CNN-XG. (3) For complex deep learning models, the current amount of data is still very small. Models such as GAN need a lot of data for generator training. We hope to explore more active learning and train better models with less data.

We hope that CNN-XG can help clinical researchers to narrow down the scope of on-target sites search and calculate the desired cleavage efficacy during the sgRNA design process, saving more time and effort for researchers.

In recent years, the number of open-source datasets for studying the application of machine learning on the CRISPR/Cas9 system has been increasing, but there is still a lack of databases to integrate existing datasets. It is believed that with the development of biotechnology, open-source datasets will gradually increase, and CNN-XG will get more comprehensive training to have better predictive abilities.

## 5. Conclusions

In this study, we present CNN-XG, an efficient hybrid model that automatically learns sequence features for CRISPR/Cas9 sgRNA activity prediction. We adopt a merged CNN architecture for sgRNA and its corresponding epigenetic feature extraction, avoiding the unknown influence of artificial feature construction processes on model prediction results, and then combine with an XGBoost classifier to predict sgRNA cleavage efficiency. Compared with CNN, XGBoost, eight advanced deep neural network models (e.g., DeepCrispr, attnToCrispr_CNN, CNN-SVR and CRISPR-ONT) and two machine learning models (e.g., SSC and sgRNA Designer), CNN-XG could efficiently utilize the data information to understand the deep features of sgRNA and the corresponding epigenetic features. Experimental results on the published datasets demonstrate the superiority of CNN-XG for CRISPR/Cas9 sgRNAs on-target activity prediction.

## Figures and Tables

**Figure 1 biomolecules-12-00409-f001:**
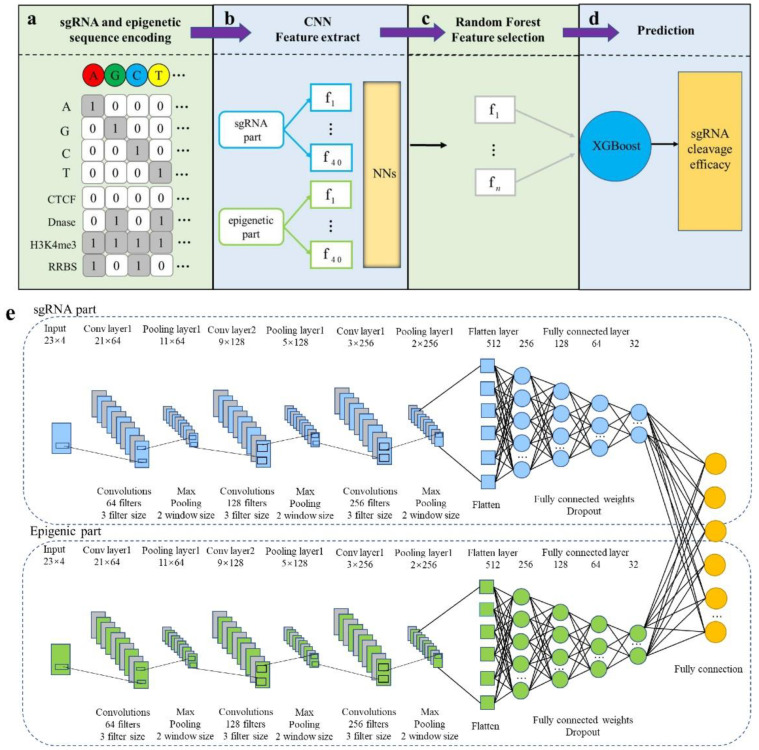
Implementation details of CNN-XG. (**a**) sgRNA and epigenetic information sequence encoding schema. There are four bases in nucleotides, A, G, C and T, each of which is seen as a channel, and each piece of epigenetic information is also seen as a channel. (**b**) Training and feature extraction in CNN. (**c**) The features extracted by the CNN are further selected using random forest models. (**d**) The selected features are put into the XGBoost classifier for the final prediction. (**e**) The structure of the convolutional part. The network contains two structurally identical branches for extracting sgRNA and epigenetic features. The final fully connected layer is used to obtain the final output.

**Figure 2 biomolecules-12-00409-f002:**
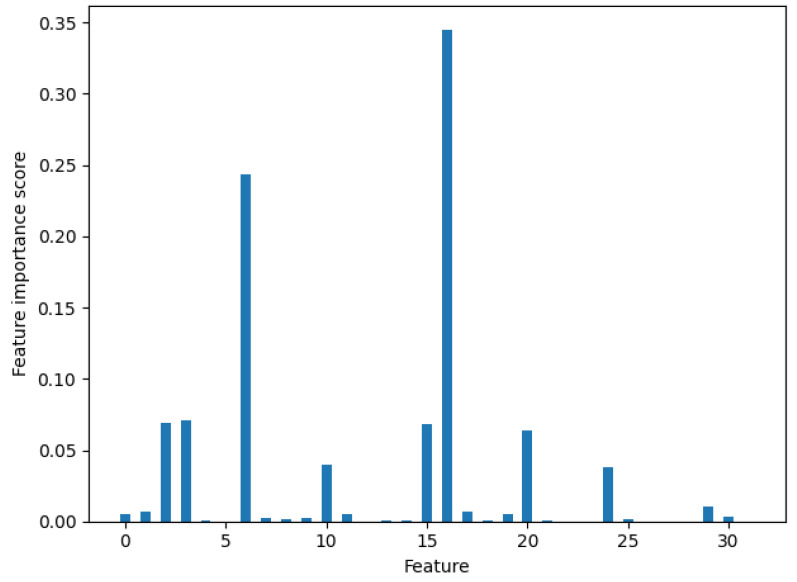
The features obtained by CNN are scored using random forest for importance (the latter 32 features are not shown because of low importance scores). X-axis represents the analyzed features, and Y-axis denotes the importance scores.

**Figure 3 biomolecules-12-00409-f003:**
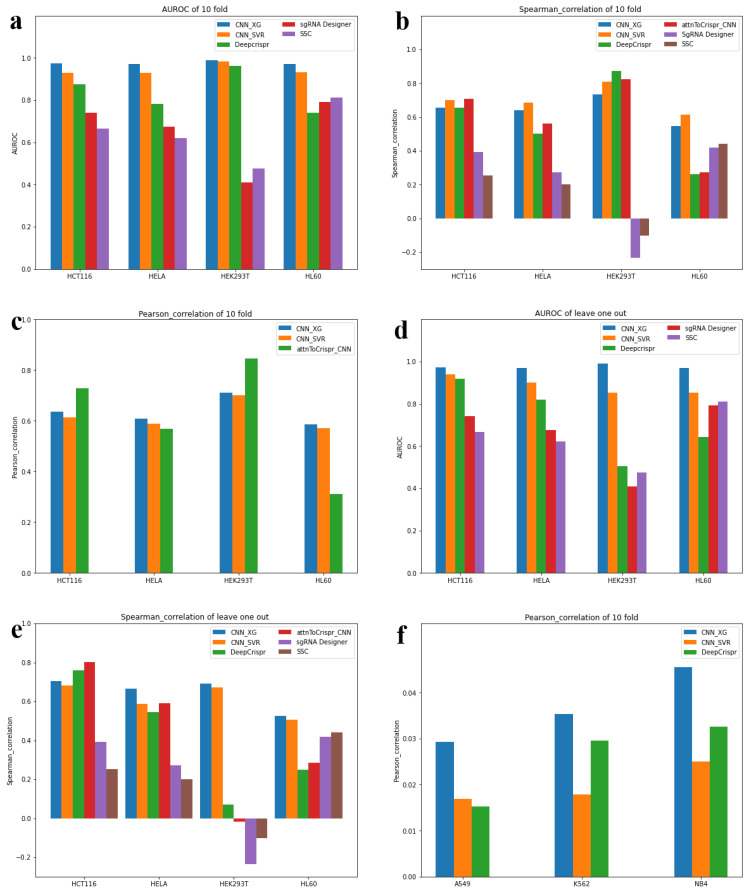
Evaluation of CNN-XG for on-target profile prediction. (**a**–**c**) Comparison of sgRNA on-target efficacy predictions for HCT116, HELA, HEK293T and HCT116 in 10-fold cross-validation in classification mode and regression mode. (**d**,**e**) Comparison of sgRNA on-target efficacy predictions in regression and classification schema for various datasets, i.e., HCT116 cell line, HELA cell line, HEK293T cell line, HCT116 cell line. (**f**) The result of a generalization ability test in new cell lines.

**Figure 4 biomolecules-12-00409-f004:**
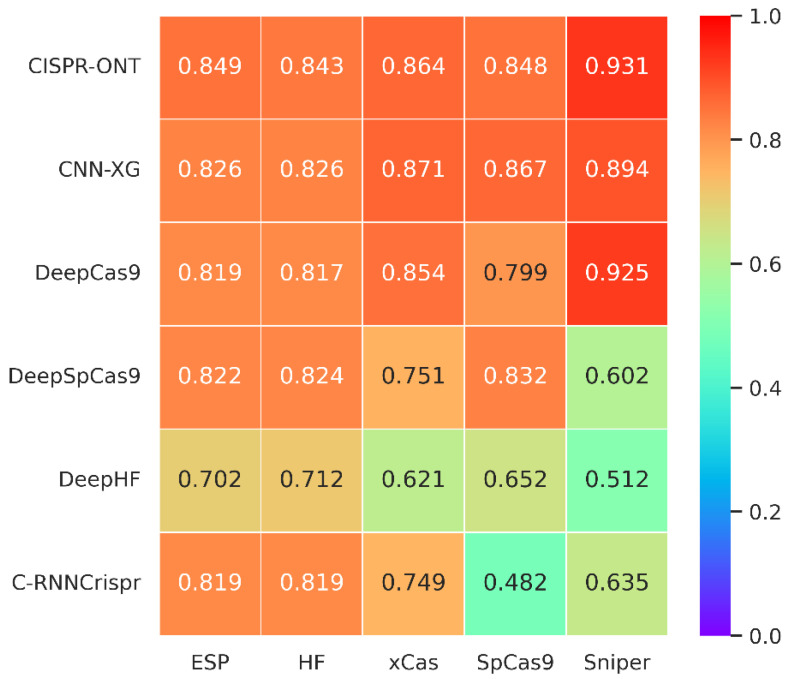
Heatmap of Spearman coefficients between CNN-XG and other recent algorithms on various SpCas9 variants datasets under 10-fold cross-validation. The prediction models are placed vertically, whereas the test sets are arranged horizontally. (The tested spearman coefficients of C-RNNCrispr, DeepCas9 and DeepSpCas9 were derived from CRISPR-ONT [34]).

**Table 1 biomolecules-12-00409-t001:** Comparison with CNN and XGBoost.

Model	CNN-XG	CNN	XGBoost	CNN-XG	CNN	XGBoost
	Spearman			AUROC		
HCT116	**0.6548**	0.6453	0.3112	**0.9732**	0.9208	0.6231
HEK293T	**0.7352**	0.7252	0.1557	**0.9905**	0.9716	0.5213
HELA	**0.6397**	0.6308	0.3273	**0.9714**	0.9163	0.6377
HL60	**0.5473**	0.5470	0.3664	**0.9706**	0.9197	0.7110

The numbers in bold indicate the highest score for each indicator.

**Table 2 biomolecules-12-00409-t002:** AUROC values for test scenario 1.

Model	CNN-XG	CNN-SVR	DeepCrispr	SgRNA Designer	SSC
HCT116	**0.9732**	0.9304	0.874	0.741	0.666
HEK293T	**0.9905**	0.9834	0.961	0.41	0.476
HELA	**0.9714**	0.9296	0.782	0.675	0.621
HL60	**0.9706**	0.9309	0.739	0.792	0.811

The numbers in bold indicate the highest score for each indicator.

**Table 3 biomolecules-12-00409-t003:** AUROC values for test scenario 2.

Model	CNN-XG	CNN-SVR	DeepCrispr	SgRNA Designer	SSC
HCT116	**0.9721**	0.939	0.919	0.741	0.666
HEK293T	**0.992**	0.852	0.506	0.41	0.476
HELA	**0.9695**	0.9024	0.82	0.675	0.621
HL60	**0.9708**	0.8526	0.643	0.792	0.811

The numbers in bold indicate the highest score for each indicator.

**Table 4 biomolecules-12-00409-t004:** Spearman correlations for test scenario 3.

Model	CNN-XG	CNN-SVR	DeepCrispr	SgRNA Designer	attnToCrispr _CNN	SSC
HCT116	0.6548	0.6998	0.654	0.391	**0.707**	0.252
HEK293T	0.7352	0.8095	**0.874**	−0.24	0.824	−0.10
HELA	0.6397	**0.6843**	0.501	0.273	0.561	0.2
HL60	0.5473	**0.6136**	0.262	0.418	0.274	0.441

The numbers in bold indicate the highest score for each indicator.

**Table 5 biomolecules-12-00409-t005:** Pearson correlations for test scenario 3.

Model	CNN-XG	CNN-SVR	AttnToCrispr_CNN
HCT116	0.6370	0.6125	**0.728**
HEK293T	0.7098	0.7000	**0.846**
HELA	**0.6079**	0.5878	0.568
HL60	**0.5849**	0.5719	0.311

The numbers in bold indicate the highest score for each indicator.

**Table 6 biomolecules-12-00409-t006:** Spearman correlations for test scenario 4.

Model	CNN-XG	CNN-SVR	DeepCrispr	SgRNA Designer	attnToCrispr _CNN	SSC
HCT116	0.703	0.683	0.761	0.391	**0.801**	0.252
HEK293T	**0.691**	0.673	0.069	−0.24	−0.02	−0.10
HELA	**0.665**	0.586	0.544	0.273	0.591	0.2
HL60	**0.527**	0.505	0.250	0.418	0.286	0.441

The numbers in bold indicate the highest score for each indicator.

**Table 7 biomolecules-12-00409-t007:** Pearson correlations and p-values on generalization test datasets.

Model	CNN-XG	CNN-SVR	DeepCrispr
A549	**0.0293** **1.405 × 10^−15^**	0.0169 4.377 × 10^−6^	0.0152 3.737 × 10^−5^
K562	**0.0353** **2.247 × 10^−30^**	0.0178 8.242 × 10^−9^	0.0295 1.002 × 10^−21^
NB4	**0.0455** **2.671 × 10^−35^**	0.0250 1.020 × 10^−11^	0.0326 6.512 × 10^−19^

The numbers in bold indicate the highest score for each indicator.

**Table 8 biomolecules-12-00409-t008:** Overall training time cost comparison of CNN-XG, CRISPR-ONT, C-RNNCrispr and DeepHF on five datasets under 10-fold cross-validation.

Model	ESP	HF	xCas	SpCas9	Sniper
CNN-XG	694	601	470	456	725
CRISPR-ONT	4123	2700	2616	2170	2640
C-RNNCrispr	161,200	15,463	10,340	9266	10,410
DeepHF	37,860	36,287	24,289	19,320	24,018

The running time of C-RNNCrispr was derived from CRISPR-ONT [34].

## Data Availability

Benchmark and four cell lines datasets used during this study are included in the published article “DeepCrispr” and its supplementary information files (DOI: https://doi.org/10.1186/s13059-018-1459-4). Three datasets for generalization ability are included in the published article “AttnToCrispr_CNN” and its supporting information files (DOI: https://doi.org/10.1371/journal.pcbi.1007480). SpCas9 variant datasets are included in the published article “DeepHF” (DOI: https://doi.org/10.1038/s41467-019-12281-8) and “DeepSpCas9” (DOI: https://doi.org/10.1038/s41587-020-0537-9). All datasets used for this study can be found at GitHub: https://github.com/MoonLBH/CNN-XG (accessed on 26 January 2022).

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
