# Peer review of "CNN-XG: A Hybrid Framework for sgRNA On-Target Prediction"

_biomolecules, 2022, doi:10.3390/biom12030409_

Round 1
Reviewer 1 Report
The authors developed a system to predict the efficacy of CRISPR/Cas9 by combining convolutional neural networks and XGBoost. The proposed system extracts features from pre-trained CNNs, and after feature selection, prediction is done by XGBoost. Comparison experiments with existing methods showed that the proposed method performed better than existing methods on data from several cell lines.
The most important feature of the authors' proposed method is that it combines CNN and XGBoost. Therefore, if you compare the results of prediction using only CNN (i.e., I think this is almost consistent with the pre-training model) and XGBoost, I think you can better highlight the merits of the proposed method.
p.1, l.28: “sgRNA”: This word is the first time it appears and needs to be spelled out.
p.1, l.36: The explanation that mismatch causes off-target is not correct, because a non-target gene can be interacted with even when no mismatch exists.
Reviewer 2 Report
The authors report a new algorithm to score sgRNA sequences for CRISPR/Cas9 experiments. Indeed, this is a field which has been widely exploited recently by a huge variety of methodological papers reporting a huge number of slightly different algorithms which attain higher and higher success in the prediction of effective sgRNA sequences. The data reported by the authors convincingly demonstrate that their approach, based on the databases used for validation, attains better results compared to other strategies.
Major points:
1) The authors omit some recent algorithms: CRISPR-ONT (PMID: 33841753), DeepHF (PMID: 31537810) which should be included in the comparative assessment of the different softwares available.
2) I could not find any detail on the computational resources required to train and run CNN-XG. The authors should include a Table comparing the running time (including model training) of the different algorithms on the same machine and discuss the outcome of this comparison in the Discussion section.
Reviewer 3 Report
Thank you for giving me an opportunity to review this paper. In this paper, authors developed a hybrid model, CNN-XG for sgRNA on-target prediction. This is indeed a good work but there are several issues that need to be addressed before considering it for publication.
- In abstract, the study objective is not clear.
- The abstract should be written clearly about objectives, methods and results.
- In the introduction, please write a clear study aim.
- In the result part: sections 3.1 and 3.2 will move to the methods part.
- Discussion: there are no clinical implications. Please extend the discussion part.
- Conclusion: It is vague. Please rewrite it clearly.
Round 2
Reviewer 1 Report
None
Reviewer 2 Report
The authors adressed all of my concerns, the manuscript can be published in its current form
Reviewer 3 Report
Thanks for your revised version. The authors have addressed all comments. Now it can be considered for publication.